# Impact of water, sanitation, and hygiene (WASH) interventions on gender-specific school attendance and learning outcomes: A systematic review and meta-analysis protocol

**Dinaol Bedada Dibaba** [1,2]*, **Bezatu Mengistie Alemu**[1]*, **Sisay Abebe Debela** [1,2]

**1** Ethiopian Institute of Water Resources, Addis Ababa University, Addis Ababa, Ethiopia, **2** Department of Public Health, College of Health Science, Salale University, Fitche, Ethiopia

* bedada.dinaol@gmail.com (DBD); bezex2000@yahoo.com (BMA)

**Data Availability Statement:** No datasets were generated or analysed during the current study. All

## Abstract

The Water, Sanitation, and Hygiene (WASH) interventions have been acknowledged for their role in the public health and educational outcomes. While there are strong evidences that reveal that WASH facilities do reduce the prevalence of infectious diseases and improve the learning environment, data remain thin and equivocal on the differential impacts of WASH facilities on education by gender. The literature reviewed does not, in most cases; go to the extent of investigating if indeed both men and women students have unique needs especially in underprivileged areas. This is the point from which the present systematic review and meta-analysis intend to fill this gap by assessing the global evidence on the effect of WASH interventions on educational outcomes with due consideration given to gender. This systematic review will include international databases used for the search, such as PubMed, Google Scholar, Web of Science, Europe PubMed Central, and Scopus. Study eligibility will include cross-sectional studies published in English on the impact of WASH interventions on school attendance and academic performance, stratifying gender-specific outcomes. Data extracted will be analyzed using the STATA software version 17. The percentage of heterogeneity will be quantified through the $I^2$ statistics to show the variability between the included studies. Based on the observed results, diversity will be checked among the outcomes of the study and based on that random-effect model will be used to estimate the pooled effect size. I will, therefore, make use of the Egger and Begg tests for checking statistical asymmetry. Publication bias will be assessed with funnel plots. These will ensure the methodologies used provide comprehensive and rigorous data analysis, which will give strong insights into the impacts of the WASH intervention on educational outcomes.

**Prospero registration number:** Systematic review and Meta-analysis registration number: PROSPERO CRD42024536477.

relevant data from this study will be made available upon study completion.

**Funding:** The author(s) received no specific funding for this work.

**Competing interests:** The authors have declared that no competing interests exist.

## 1. Introduction

Water, Sanitation, and Hygiene (WASH) interventions are key for public health and have a big impact on education and social outcomes all over the world [1, 2]. Schools especially need good WASH facilities not just for basic cleanliness but also because they help kids come to school more often and boost their ability to learn [3]. There is evidence that having the right sanitation facilities at school can make a real difference. Schools that have them see their students doing better in their studies compared to schools without them [10, 11]. This probably comes down to fewer kids getting sick and a better overall environment at school [2, 12].

It is very important to have effective WASH practices, particularly in places where resources are scarce. Without these, there can be major health and learning problems. Studies have shown that poor WASH setups lead to more sickness and lower school performance, especially in less developed areas [4, 5]. Take diarrheal diseases, for instance. They are a big reason why kids miss school worldwide, and better sanitation could help reduce this issue [1, 6]. In addition, not having proper and safe toilets really affects whether kids especially girls go to school. Girls need these facilities to manage menstrual hygiene, and when they are not there, it hits girls attendance hard [7–9].

On a larger scale, setting up WASH programs in schools contributes to wider economic and social benefits. It helps level the educational playing field and moves society forward [13, 14]. These efforts are also part of reaching the United Nations' Sustainable Development Goals, which include targets for health, education, and gender equality [15, 16]. But, despite these benefits, the challenge remains huge. Millions of schools around the world still do not have basic WASH facilities, affecting underserved and rural areas hardest and continuing cycles of poverty and poor health [17], [18]. Addressing this issue needs a well-rounded approach that includes installing facilities, educating on hygiene, and getting the community involved [19, 20]. Adapting efforts to fit the unique cultural and social contexts of communities can make WASH programs more effective [21].

Despite the recognized benefits of Water, Sanitation, and Hygiene (WASH) interventions, critical gaps exist, particularly in enabling a gender-oriented assessment of their impacts. Most available studies lack gender sensitivity or consideration of socio-cultural factors, which significantly influence the effectiveness of these interventions [22]. Additionally, the variability of outcomes across different geographic and socio-economic contexts, influenced by local cultural norms and educational policies, complicates the formulation of effective strategies [1, 23]. This meta-analysis aims to synthesize global research to elucidate how WASH interventions impact educational outcomes differently for boys and girls, providing precise, actionable insights that can inform policy and enhance educational equity through tailored WASH practices.

## 2. Review question

1. How do WASH interventions influence school attendance rates, specifically analyzing differences based on gender?

2. How do WASH interventions impact academic performance, as measured by standardized tests or other indicators, for boys and girls?

3. Do the effects of WASH interventions on educational outcomes demonstrate variability across different geographic regions?

## 3. Methodology

### 3.1. Search strategy

For this systematic review and meta-analysis, different search strategies were employed including but not limited to general search string and database-specific search string such as PubMed, PubMed Central, Scopus, Web of Science and Cochrane Library. Additionally, we will include a backward citation tracking approach as part of our search strategy. This involves reviewing the reference lists of all included studies to identify any additional studies that may not have been captured in our initial database searches. This comprehensive approach will help ensure that we include all relevant studies in our review, thereby enhancing the completeness of our systematic review and meta-analysis.

**3.1.1 General search string.** ("water, sanitation, and hygiene" OR "WASH interventions" OR "sanitation facilities" OR "hygiene facilities") AND ("school attendance" OR "academic performance" OR "learning outcomes" OR "educational outcomes").

**3.1.2. Database-specific search strings.**

**a) PubMed**

For PubMed free text and Mesh terms, (((((((((("impact"[Title/Abstract] OR "effect"[Title/Abstract]) AND "water"[Title/Abstract]) OR "water"[MeSH Terms]) AND "sanitation"[Title/Abstract]) OR "sanitation"[MeSH Terms]) AND "hygiene"[Title/Abstract]) OR "hygiene"[MeSH Terms]) AND "School"[Title/Abstract]) OR "schools"[MeSH Terms]) AND "attendance"[Title/Abstract] search queries were used.

**PubMed free text searches**, ((((impact[tiab]) OR (effect[tiab])) AND ((((water[tiab]) AND (sanitation[tiab])) AND (hygiene[tiab])) OR (WASH[tiab]))) AND (School[tiab])) AND (attendance[tiab]) search queries were used.

**b) Scopus**

For Scopus which allows more flexible text word searches, TITLE-ABS-KEY ("Impact of water, sanitation, and hygiene" OR "effect of water, sanitation, and hygiene" OR "effect of WASH" OR "impact of WASH" OR "effect of sanitation facilities" OR "impact of sanitation facilities" OR "effect of hygiene" OR "impact of hygiene") AND ("school attendance" OR "academic performance" OR "learning outcomes") search queries were used.

**c) Web of Science**

Web of Science search interface using Boolean operators and nested queries, TS = ((" water, sanitation, and hygiene" OR " WASH interventions" OR " sanitation facilities" OR "impact of hygiene facilities") AND (" school attendance" OR "academic performance" OR "learning outcomes")) search queries were used.

**d) Cochrane Library**

For Cochrane Library, ("water, sanitation, and hygiene" OR "WASH interventions" OR "sanitation" OR "hygiene") AND ("school attendance" OR "learning outcomes" OR "academic achievement") search queries were used.

### 3.2 Screening strategy

Titles and abstracts will be independently screened to assess potential relevance based on the inclusion and exclusion criteria.

### 3.3. Inclusion and exclusion criteria

#### 3.3.1. Inclusion criteria.

**Study Types:** Institutional cross-sectional studies, Controlled Trials (RCTs), causal-comparative studies, matched-control trials, and case-control studies.

**Population:** School-aged children (ages 6–18) attending formal educational institutions.

**Interventions:** Studies evaluating the impact of WASH interventions.

**Outcomes:** Studies must report gender-specific educational outcomes such as differences in school attendance or academic performance between male and female students.

**Data source:** Both published and unpublished studies done in English available globally will be considered.

#### 3.3.2. Exclusion criteria.
Studies that do not provide data disaggregated by gender, and studies not in educational settings.

### 3.4. Data extraction and reporting

To ensure uniformity and free from biases when extracting data from all of the studies included in this review, We will use standardized data extraction forms. The form has been designed to capture details of information that are critical in understanding the scope and impacts of WASH interventions. From each of the studies, We will extract the following:

**Author(s) and year of publication:** For citation and reference tracking.

**Country of study:** To evaluate geographical diversity and contextual applicability.

**Study design:** This encompasses the design of the studies undertaken.

**Population details:** Demographic description of the study population, such as the age and gender distribution.

**Education outcome:** Include data for schooling attendance of education outcome and academic achievement outcome, paying special focus on gender disparity.

**Sample size:** Number of participants, which is critical for statistical analyses. Statistical **Measures:** Key statistics such as effect sizes, confidence intervals, and p-values.

### 3.5. Data analysis

To ascertain the level of heterogeneity among the included studies on the impact of Water, Sanitation, and Hygiene (WASH) interventions on gender-specific educational outcomes, the $I^2$ statistic will be utilized, with values over 75% indicating substantial heterogeneity. This is a common approach in educational and social sciences meta-analysis [24, 25]. All statistical analyses will be rigorously conducted using STATA version 17, facilitating a comprehensive and precise synthesis of data aimed at enhancing educational and public health strategies.

We will use different measures of effect sizes depending on the type of outcome data. For continuous outcomes, we will calculate the standardized mean difference (SMD), and for dichotomous outcomes, we will use the risk ratio (RR) or odds ratio (OR). These effect size measures provide a standardized way to compare results across studies with different scales and outcome measures. To calculate the pooled effect size, we will use a random-effects model,

which accounts for both within-study and between-study variability. This model is particularly appropriate when there is substantial heterogeneity among the studies.

We will conduct subgroup analyses and meta-regressions to explore potential sources of heterogeneity between studies, in addition to using the $I^2$ statistic. Subgroup analyses will compare the effects of WASH interventions across different genders and geographic regions. Meta-regressions will include covariates such as socio-economic status, urban versus rural settings, and baseline WASH conditions to control for potential confounders.

### 3.6. Sensitivity analysis and certainty assessment of evidence

In this protocol, sensitivity analysis is systematically applied to ensure the reliability of the findings regarding the effects of Water, Sanitation and Hygiene (WASH) interventions on educational outcomes. This methodological approach involves testing the stability of the results under various conditions, such as using different statistical models (fixed-effect vs. random-effect) and altering the inclusion criteria to exclude studies with higher risks of bias or lower quality. By doing so, We aim to identify whether the conclusions drawn are sensitive to the chosen methods or specific subsets of data. This thorough examination helps validate the robustness of the outcomes, enhancing the study's credibility and providing clear, evidence-based insights that can guide policy-making and educational program enhancements. Additionally, sensitivity analysis promotes transparency in the research process, allowing for a deeper understanding of how methodological choices influences the study's conclusions.

We will assess the certainty of the evidence using the GRADE (Grading of Recommendations, Assessment, Development, and Evaluations) tool. This approach will allow us to systematically evaluate the quality of evidence and provide transparent and reliable conclusions about the impact of WASH interventions on educational outcomes.

### 3.7. Critical appraisal

Critical Appraisal will be systematically employed to ensure the integrity and validity of the included studies. This will be encouraged through the application of standard tools, such as Newcastle-Ottawa Scale for the observational studies.

To assess the quality of unpublished studies, first, we will use established quality appraisal tools, such as the Joanna Briggs Institute (JBI) checklist, to evaluate all studies regardless of publication status. This ensures that only studies meeting a minimum quality threshold are included. Additionally, two independent reviewers will assess each study's quality, resolving discrepancies through discussion or a third reviewer if necessary. We will conduct sensitivity analyses to evaluate the impact of including unpublished studies on our findings, ensuring the robustness of our results. Furthermore, we will assess publication bias using funnel plots and Egger's regression test to identify and mitigate any biases. These techniques are well-regarded for their efficacy in detecting asymmetry, which may indicate the presence of publication bias.

### 3.8 Potential limitations of this review

Several potential limitations are anticipated in this meta-analysis. One of the concerns is the variability in how WASH interventions are implemented across different settings, which might affect the comparability of results. Additionally, there is a risk of publication bias, as studies showing significant or positive results are more likely to be published than those that do not, potentially skewing the overall findings. Given the diverse settings of the studies involved, there may also be substantial heterogeneity, which could complicate the interpretation of the pooled estimates. In some cases, relevant data may be incomplete or missing, limiting the analysis of certain outcomes. Finally, the impact of socio-cultural variables, which

could influence the effectiveness of WASH interventions, might be underrepresented due to the lack of detailed reporting in the original studies.

## 3.9 Operational definitions

**WASH interventions:** Initiatives aimed at improving water quality, sanitation facilities, and hygiene practices within school settings. These interventions are intended to enhance public health and educational outcomes by providing cleaner environments and reducing disease transmission.

**School attendance:** The regular presence of school-aged children (ages 6–18) in formal educational institutions.

**Academic performance:** The educational outcomes and achievements of students in their schoolwork.

**Learning outcomes:** While term summarizes the broader cognitive, emotional, and social competencies acquired by students through schooling. Learning outcomes in this document are evaluated in relation to the presence and quality of WASH facilities.

**Gender-specific educational outcomes:** These are the differences in educational results between male and female students, particularly focusing on how WASH interventions may affect each gender differently, such as through facilities that cater to menstrual hygiene management that primarily affects female students.

**Statistical measures (effect sizes, confidence intervals, p-values):** Used to describe the magnitude and significance of the effects observed in the study. These include the size of the impact WASH interventions have on gender-specific educational outcomes, the reliability of these estimates, and the statistical significance of the results.

**Heterogeneity ($I^2$ statistic):** A statistical measure used to quantify the variance among the included studies. Values over 75% indicate substantial heterogeneity, affecting the interpretation of pooled data in meta-analyses.

**Publication bias (funnel plots, egger's regression test):** Techniques used to assess asymmetry in meta-analysis data, which can indicate whether the synthesis of results might be biased by the non-publication of negative or inconclusive studies.

## 3.10. Quality assessment and study selection process

For the quality assessment of the studies included in our systematic review and meta-analysis on the effects of Water Sanitation and Hygiene (WASH) interventions on educational outcomes, two authors (Dinaol Bedada and Bezatu Mengistie) will independently evaluate each study. The quality of the included studies will be assessed using the Joanna Briggs Institute (JBI) critical appraisal checklist. This tool is well-regarded for its comprehensive approach to evaluating the methodological quality of studies across various designs. [26, 27]. Each study will be scored on a scale from 0 to 100%, and only those receiving a score of 50% or higher will be included in further analysis [28]. In cases where there are discrepancies between the assessments, DBD and BMA will discuss to resolve differences or, if necessary, involve a third, impartial reviewer to achieve consensus. The results of this quality assessment is indicated in a PRISMA flow diagram (Fig 1) [29], which will detail the study selection process, including the number of studies screened, assessed for eligibility, included in the review, and excluded with

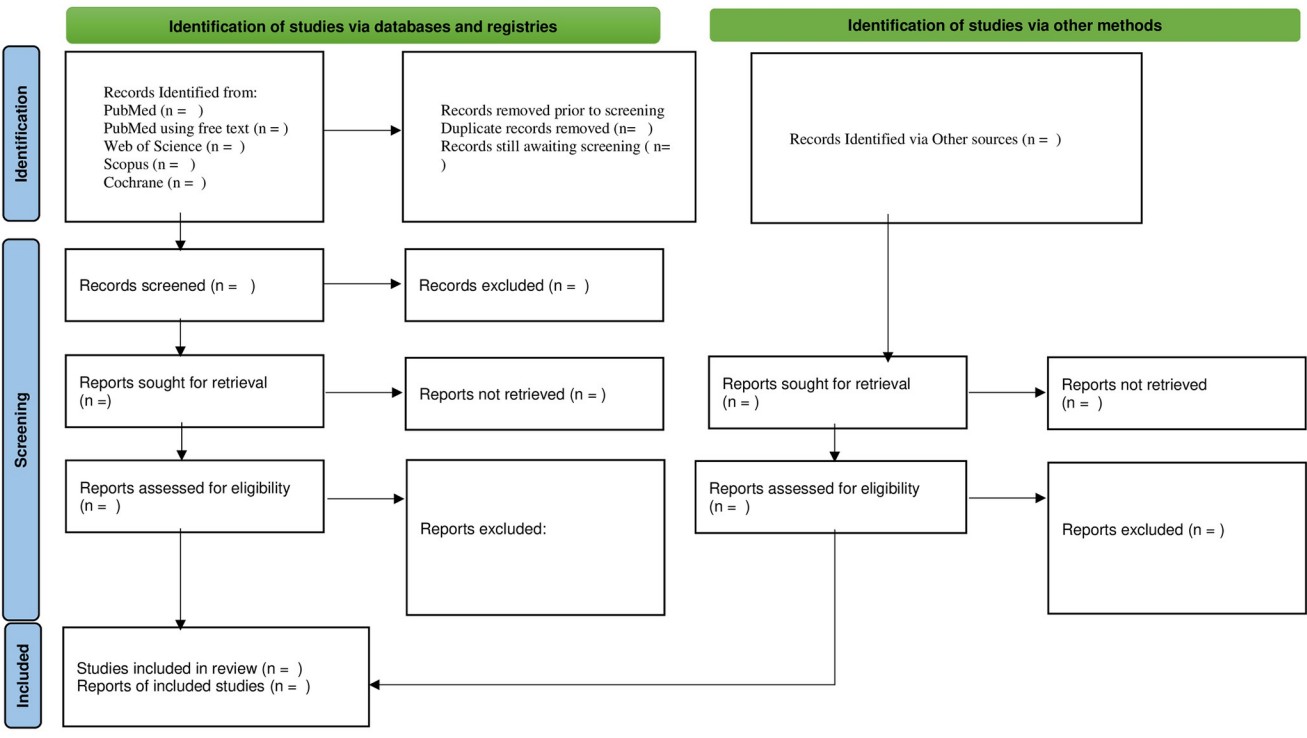

**Fig 1. PRISMA Diagram for the study selection.**

reasons at each stage. This approach ensures the reliability and validity of the included studies in our meta-analysis.

To prevent counting multiple publications from the same research project as separate studies, we will screen for duplicates during data extraction by checking author names, study details, and periods. If identified, we will consolidate the data, using the most comprehensive version. We will document these instances and our approach in the final report for transparency.

**3.1.1. Bias risk assessment.**   The risk of bias will be assessed by evaluating heterogeneity using the $I^2$ statistic. Additionally, publication bias will be evaluated using Egger's regression test and funnel plots. These methods will help identify and quantify any potential bias in the studies included in the meta-analysis.

**3.1.2. Data extraction form.**   The data extraction form captures key information necessary for our analysis. This includes study characteristics such as author(s), year of publication, country, study design, and sample size. It also details data on WASH facilities and school attendance, distinguishing between students attending schools with adequate WASH facilities and those not attending despite such facilities. Additionally, the form collects data on academic performance and learning outcomes, including test scores, grades, and other measurable educational achievements. Gender-specific data on both school attendance and academic performance will also be recorded to analyze differential impacts. To ensure accuracy, two independent reviewers will extract the data, resolving discrepancies through discussion or involving a third reviewer. See "Data extraction form–DINAOL.docx" file for details.

## 4. Discussion

Significant gaps exist in the current literature regarding the differential impacts of Water Sanitation and Hygiene (WASH) interventions on educational outcomes for students, particularly in disadvantaged regions. These disparities often go unaddressed, highlighting the need for a systematic review and meta-analysis. By focusing on gender-specific educational outcomes, this systematic review and Meta-Analysis aims to provide a detailed understanding of how these interventions can meet the unique needs of different student groups, thereby enhancing their effectiveness and inclusivity.

Our research approach involved search strategy across several databases to ensure a thorough evaluation of available data. We executed detailed searches in PubMed, Web of Science, Scopus, and the Cochrane Library, retrieving a many records: 1677 from PubMed, 27 from Web of Science, 7 from Scopus, and 67 from the Cochrane Library. Our strategy employed both free text and controlled vocabulary terms to maximize the capture of relevant studies, highlighting the scale and depth of our investigation. This methodological approach is intended to capture a wide range of studies, ensuring a detailed analysis of the available evidence.

In synthesizing this evidence, we expect to fill the significant research gaps by providing robust, differentiated insights into the effectiveness of WASH interventions. This will not only contribute to the academic literature but also offer actionable guidance for policymakers and educational authorities. By contextualizing the potential outcomes within the scope of public health and educational enhancement, we aim to foster a more inclusive and effective approach to implementing WASH programs globally. This protocol sets the stage for a comprehensive exploration of WASH interventions, paving the way for future studies and interventions that are informed by gender-sensitive research and practice.

## Supporting information

**S1 Checklist.**
(DOCX)

**S1 Data.**
(DOCX)

**S1 File.**
(DOCX)

**S2 File.**
(DOCX)

## Author Contributions

**Conceptualization:** Dinaol Bedada Dibaba, Bezatu Mengistie Alemu.

**Data curation:** Dinaol Bedada Dibaba, Bezatu Mengistie Alemu, Sisay Abebe Debela.

**Formal analysis:** Dinaol Bedada Dibaba, Bezatu Mengistie Alemu, Sisay Abebe Debela.

**Funding acquisition:** Dinaol Bedada Dibaba.

**Investigation:** Dinaol Bedada Dibaba.

**Methodology:** Dinaol Bedada Dibaba, Bezatu Mengistie Alemu, Sisay Abebe Debela.

**Project administration:** Dinaol Bedada Dibaba.

**Resources:** Dinaol Bedada Dibaba, Sisay Abebe Debela.

**Software:** Dinaol Bedada Dibaba.

**Supervision:** Dinaol Bedada Dibaba, Bezatu Mengistie Alemu.

**Validation:** Dinaol Bedada Dibaba, Sisay Abebe Debela.

**Visualization:** Dinaol Bedada Dibaba.

**Writing – original draft:** Dinaol Bedada Dibaba.

**Writing – review & editing:** Dinaol Bedada Dibaba, Bezatu Mengistie Alemu, Sisay Abebe Debela.

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
