## [Decision Letter · Decision Letter 0]

20 Jun 2024

PONE-D-24-18212Impact of Water, Sanitation, and Hygiene (WASH) Interventions on Gender-Specific School Attendance and Learning Outcomes: A Systematic Review and Meta-Analysis ProtocolPLOS ONE

Dear Dr. Dibaba,

Thank you for submitting your manuscript to PLOS ONE. After careful consideration, we feel that it has merit but does not fully meet PLOS ONE’s publication criteria as it currently stands. Therefore, we invite you to submit a revised version of the manuscript that addresses the points raised during the review process.

Please submit your revised manuscript by Aug 04 2024 11:59PM. If you will need more time than this to complete your revisions, please reply to this message or contact the journal office at plosone@plos.org. Please include the following items when submitting your revised manuscript:A rebuttal letter that responds to each point raised by the academic editor and reviewer(s). You should upload this letter as a separate file labeled 'Response to Reviewers'.A marked-up copy of your manuscript that highlights changes made to the original version. You should upload this as a separate file labeled 'Revised Manuscript with Track Changes'.An unmarked version of your revised paper without tracked changes. You should upload this as a separate file labeled 'Manuscript'.

We look forward to receiving your revised manuscript.

Kind regards,

Ashish Wasudeo Khobragade, MD

Academic Editor

PLOS ONE

Journal Requirements:

Additional Editor Comments:

The topic is interesting. The following issues are not clear after reading the protocol.

1. Cite the reference number instead of the author's name and year in the introduction section.

2. Which quality assessment tool will be used?

3. How risk of bias will be assessed?

4. Mention in detail how statistical analysis will be done. What are the different measures of effect sizes that will be used? Which statistical method will be used to calculate the pooled effect size?

Reviewers' comments:

Reviewer's Responses to Questions

**Comments to the Author**

1. Does the manuscript provide a valid rationale for the proposed study, with clearly identified and justified research questions?

Reviewer #1: Yes

Reviewer #2: Yes

2. Is the protocol technically sound and planned in a manner that will lead to a meaningful outcome and allow testing the stated hypotheses?

Reviewer #1: Yes

Reviewer #2: Yes

3. Is the methodology feasible and described in sufficient detail to allow the work to be replicable?

Reviewer #1: Yes

Reviewer #2: Yes

4. Have the authors described where all data underlying the findings will be made available when the study is complete?

Reviewer #1: No

Reviewer #2: Yes

5. Is the manuscript presented in an intelligible fashion and written in standard English?

Reviewer #1: No

Reviewer #2: Yes

6. Review Comments to the Author

You may also provide optional suggestions and comments to authors that they might find helpful in planning their study.

**Reviewer #1: **Overall, the protocol appears to be well-designed and clearly outlines the planned systematic review and meta-analysis on the impact of WASH interventions on gender-specific school attendance and learning outcomes. The methodology section provides sufficient detail on the planned search strategy, study eligibility criteria, data extraction, and statistical analysis.

A few suggestions that could potentially strengthen the protocol:

Consider including a section on the planned approach for assessing the quality and risk of bias of the included studies. This is an important step in a systematic review.

Specify if the authors will conduct subgroup analyses or meta-regressions to explore potential sources of heterogeneity between studies, in addition to the planned use of I^2 statistic.

Clarify if the authors will assess the certainty of the evidence using a tool like GRADE.

Consider discussing any potential limitations or challenges that may be encountered during the conduct of this systematic review.

**Reviewer #2:** WASH interventions have important public health implications. They can improve the health and wellbeing of individuals and communities, and can help improve school attendance and learning outcomes. The present study attempts to examine the impact of WASH across different genders and geographic regions. Some specific comments:

1. The search strategies are clearly and explicitly defined. However the reference lists of the selected papers can also serve as a source for identifying additional relevant studies to be included in the present review.

2. It was stated that the data source includes both published and unpublished studies (line 121). However unpublished studies may not have undergone the same level of peer review and quality control as published studies, which may make it difficult to assess the validity and reliability of the findings, and hence the quality of these studies may be uncertain. How do you determine the inclusion of these studies?

3. Some research projects may have published multiple journal papers. It would not be appropriate to count them as separate and studies. Please consider how it can be avoided.

4. It seems that the present review only focuses on cross-sectional studies (line 116). Other study designs, such as RCT, will provide stronger evidence on causal relationship between WASH interventions and educational outcomes. It is possible that schools with WASH interventions are more likely to be found in higher socio-economic areas or better-resourced schools that can afford such improvements. In these cases, educational outcomes may be due to higher socio-economic status, rather than WASH interventions. Please explain why only cross-sectional studies will be included.

5. The statistical methods (line 139) are appropriate, but need detailed analysis strategies used to address the impact of WASH interventions across different genders and different geographic regions.

6. The data extraction form (line 217) (Data extraction form --DINAOL.docx) is not clear. For example, what the meanings of “WASH available and attend school” and “WASH available and not attend school” are. There are no mentions of how data on academic performance and learning outcomes are extracted. The lack of detail on data collection methods may affect the validity and reliability of the findings.

7. PLOS authors have the option to publish the peer review history of their article (what does this mean?). If published, this will include your full peer review and any attached files.

Reviewer #1: **Yes: **NUHA AMER Al-Aghbari

Reviewer #2: **Yes: **LM Ho

---

## [Author Response · Author response to Decision Letter 0]

24 Jun 2024

In addition to the following response, we uploaded the Response to the reviewers In separate file.

Response to reviewers

I. Additional Editor Comments and Authors Responses:

1.Cite the reference number instead of the author's name and year in the introduction section.

Authors Response: Thank you for highlighting this. The referencing style has been changed to IEEE format, which uses numbered citations to comply with the journal's guidelines.

2.Which quality assessment tool will be used?

Authors Response: Thank you for mentioning this constructive question. The quality of the included studies will be assessed using the Joanna Briggs Institute (JBI) critical appraisal checklist. This tool is well-regarded for its comprehensive approach to evaluating the methodological quality of studies across various designs. The modification is reflected in the protocol sections of quality assessment (4.10).

3.How risk of bias will be assessed?

Authors Response: The risk of bias will be assessed by evaluating heterogeneity using the I² statistic. Additionally, publication bias will be evaluated using Egger's regression test and funnel plots. These methods will help identify and quantify any potential bias in the studies included in the meta-analysis. The protocol is modified accordingly, Section 4.11, Bias Risk Assessment. 

4.Mention in detail how statistical analysis will be done. What are the different measures of effect sizes that will be used? Which statistical method will be used to calculate the pooled effect size?

Authors Response: Thank you for highlighting this concern. To ensure a strong and comprehensive statistical analysis, we will employ several detailed methods. First, we will assess heterogeneity among the included studies using the I² statistic, with values over 75% indicating substantial heterogeneity. This measure helps us understand the degree of variability in effect estimates due to heterogeneity rather than chance.

We will use different measures of effect sizes depending on the type of outcome data. For continuous outcomes, we will calculate the standardized mean difference (SMD), and for dichotomous outcomes, we will use the risk ratio (RR) or odds ratio (OR). These effect size measures provide a standardized way to compare results across studies with different scales and outcome measures.

To calculate the pooled effect size, we will use a random-effects model, which accounts for both within-study and between-study variability. This model is particularly appropriate when there is substantial heterogeneity among the studies. Publication bias will be evaluated through Egger’s regression test and the visual examination of funnel plots. Statistical analyses will be conducted using STATA version 17. This software facilitates a comprehensive and precise synthesis of data, enhancing the reliability of our findings. Our protocol is updated accordingly (Section 4.5 Data Analysis).

II. Reviewers Comments and Authors Responses:

Reviewer #1: 

Overall, the protocol appears to be well-designed and clearly outlines the planned systematic review and meta-analysis on the impact of WASH interventions on gender-specific school attendance and learning outcomes. The methodology section provides sufficient detail on the planned search strategy, study eligibility criteria, data extraction, and statistical analysis.

A few suggestions that could potentially strengthen the protocol:

1.Consider including a section on the planned approach for assessing the quality and risk of bias of the included studies. This is an important step in a systematic review.

Authors Response: Thank you for your constructive feedback and suggestions to strengthen our protocol. We have included a section (4.11. Bias Risk Assessment) detailing our approach for assessing the quality and risk of bias of the included studies. We will use the Joanna Briggs Institute (JBI) critical appraisal checklist for this purpose. This tool is well-regarded for evaluating the methodological quality of studies across various designs.

2. Specify if the authors will conduct subgroup analyses or meta-regressions to explore potential sources of heterogeneity between studies, in addition to the planned use of I2 statistic.

Authors Response: Thank you for your constructive suggestions. We will conduct subgroup analyses and meta-regressions to explore potential sources of heterogeneity between studies, in addition to using the I² statistic. Subgroup analyses will compare the effects of WASH interventions across different genders and geographic regions. Meta-regressions will include covariates such as socio-economic status, urban versus rural settings, and baseline WASH conditions to control for potential confounders. The modification is reflected in the protocol sections of 4.5 data Analysis.

3.Clarify if the authors will assess the certainty of the evidence using a tool like GRADE.

Authors Response: Thank you for your constructive implication. We will assess the certainty of the evidence using the GRADE (Grading of Recommendations, Assessment, Development, and Evaluations) tool. This approach will allow us to systematically evaluate the quality of evidence and provide transparent and reliable conclusions about the impact of WASH interventions on educational outcomes. The modification is reflected in the protocol sections of 4.6. Sensitivity Analysis and Certainty Assessment of Evidence.

4.Consider discussing any potential limitations or challenges that may be encountered during the conduct of this systematic review.

Authors Response: Thank you for your constructive suggestion. The potential limitations and challenges that may be encountered during the conduct of this systematic review is indicated under limitations section of the protocol (4.8 limitation of Review). These include variability in study quality, differences in WASH intervention implementation, potential publication bias, and the challenge of synthesizing data from diverse geographic and socio-economic contexts. By acknowledging these limitations, we aim to provide a balanced interpretation of our findings.

Reviewer #2: 

WASH interventions have important public health implications. They can improve the health and wellbeing of individuals and communities, and can help improve school attendance and learning outcomes. The present study attempts to examine the impact of WASH across different genders and geographic regions. Some specific comments:

1. The search strategies are clearly and explicitly defined. However, the reference lists of the selected papers can also serve as a source for identifying additional relevant studies to be included in the present review.

Authors Response: Thank you for your constructive comments and suggestions. We appreciate your recognition of the importance of WASH interventions and the clarity of our search strategies. We agree that reference lists of selected papers can be valuable sources for identifying additional relevant studies. Therefore, we will include a backward citation tracking approach as part of our search strategy. This involves reviewing the reference lists of all included studies to identify any additional studies that may not have been captured in our initial database searches. This approach will help ensure that we include all relevant studies in our review, thereby enhancing the completeness of our systematic review and meta-analysis. This indicated in 4.1 Search Strategy section of the protocol.

2. It was stated that the data source includes both published and unpublished studies (line 121). However unpublished studies may not have undergone the same level of peer review and quality control as published studies, which may make it difficult to assess the validity and reliability of the findings, and hence the quality of these studies may be uncertain. How do you determine the inclusion of these studies?

Authors Response: Thank you for your valuable feedback. We acknowledge the concern regarding the inclusion of unpublished studies and the potential challenges in assessing their quality. To address this, we will implement several measures. First, we will use established quality appraisal tools, such as the Joanna Briggs Institute (JBI) checklist, to evaluate all studies regardless of publication status. This ensures that only studies meeting a minimum quality threshold are included. Additionally, two independent reviewers will assess each study's quality, resolving discrepancies through discussion or a third reviewer if necessary. We will conduct sensitivity analyses to evaluate the impact of including unpublished studies on our findings, ensuring the strength of our results. Furthermore, we will assess publication bias using funnel plots and Egger’s regression test to identify and mitigate any biases. These modifications are reflected in the protocol sections of critical appraisal (4.7 critical appraisal).

3. Some research projects may have published multiple journal papers. It would not be appropriate to count them as separate and studies. Please consider how it can be avoided.

Authors Response: Thank you for highlighting this concern. To prevent counting multiple publications from the same research project as separate studies, we will screen for duplicates during data extraction by checking author names, study details, and periods. If identified, we will consolidate the data, using the most comprehensive version. We will document these instances and our approach in the final report for transparency. This indicated in the last paragraph of section 4.10. of the protocol (4.10. Quality assessment and study selection process).

4. It seems that the present review only focuses on cross-sectional studies (line 116). Other study designs, such as RCT, will provide stronger evidence on causal relationship between WASH interventions and educational outcomes. It is possible that schools with WASH interventions are more likely to be found in higher socio-economic areas or better-resourced schools that can afford such improvements. In these cases, educational outcomes may be due to higher socio-economic status, rather than WASH interventions. Please explain why only cross-sectional studies will be included.

Authors Response: Thank you for highlighting this concern. We acknowledge the limitation of focusing solely on cross-sectional studies, which may not adequately address the potential confounding effect of socio-economic status. To strengthen the evidence on the causal relationship between WASH interventions and educational outcomes, we have expanded our inclusion criteria to encompass various study designs. These include Randomized Controlled Trials (RCTs), causal-comparative studies, matched-control trials, and case-control studies.

By including these diverse study designs, we aim to mitigate the risk of confounding factors such as socio-economic status, ensuring a more robust and comprehensive analysis. This approach allows us to account for the potential that schools with WASH interventions may also have other resources that contribute to better educational outcomes. The modifications are indicated in the updated protocol, which reflects our commitment to including higher-quality evidence to support our findings (Section 4.3.1 inclusion criteria)

5. The statistical methods (line 139) are appropriate, but need detailed analysis strategies used to address the impact of WASH interventions across different genders and different geographic regions.

Authors Response: Thank you for your valuable feedback. We have enhanced our analysis strategies to address the impact of WASH interventions across different genders and geographic regions. Specifically, we will perform subgroup analyses and include interaction terms in our meta-regression models to compare the effects on boys and girls. Additionally, we will conduct geographic-specific subgroup analyses and use meta-regression to explore how location moderates the impact of WASH interventions, controlling for covariates like socio-economic status and urban versus rural settings. These strategies, included in our updated protocol (4.5. Data Analysis), will ensure a comprehensive and strong analysis of the varied impacts of WASH interventions.

6. The data extraction form (line 217) (Data extraction form --DINAOL.docx) is not clear. For example, what the meanings of “WASH available and attend school” and “WASH available and not attend school” are. There are no mentions of how data on academic performance and learning outcomes are extracted. The lack of detail on data collection methods may affect the validity and reliability of the findings.

 Authors Response: Thank you for highlighting these concerns. We have revised our protocol to provide clearer definitions and more detailed data collection methods to ensure the validity and reliability of our findings.

We have clarified the terms used in the data extraction form. “WASH available and attend school” refers to the number of students attending school where adequate WASH facilities are present. Conversely, “WASH available and not attend school” refers to the number of students not attending school despite the presence of adequate WASH facilities. These distinctions help us accurately capture the relationship between WASH facilities and school attendance.

Secondly, we have included the process for extracting data on academic performance and learning outcomes. This includes collecting information on test scores, grades, and other measurable educational achievements from each study. These data points will be categorized and analyzed to assess the impact of WASH interventions on educational outcomes.

To ensure accuracy, two independent reviewers will perform the data extraction, with discrepancies resolved through discussion or by involving a third reviewer. These updates are reflected in our revised protocol (Section 4.12 Data Extraction form), and in our data extraction form (“Data extraction form --DINAOL.docx”) ensuring a comprehensive and reliable data extraction process.

---

## [Decision Letter · Decision Letter 1]

18 Jul 2024

Impact of Water, Sanitation, and Hygiene (WASH) Interventions on Gender-Specific School Attendance and Learning Outcomes: A Systematic Review and Meta-Analysis Protocol

PONE-D-24-18212R1

Dear Dr. Dibaba,

We’re pleased to inform you that your manuscript has been judged scientifically suitable for publication and will be formally accepted for publication once it meets all outstanding technical requirements.

Kind regards,

Ashish Wasudeo Khobragade, MD

Academic Editor

PLOS ONE

Additional Editor Comments (optional):

Reviewers' comments:

Reviewer's Responses to Questions

**Comments to the Author**

1. Does the manuscript provide a valid rationale for the proposed study, with clearly identified and justified research questions?

Reviewer #2: Yes

2. Is the protocol technically sound and planned in a manner that will lead to a meaningful outcome and allow testing the stated hypotheses?

Reviewer #2: Yes

3. Is the methodology feasible and described in sufficient detail to allow the work to be replicable?

Reviewer #2: Yes

4. Have the authors described where all data underlying the findings will be made available when the study is complete?

Reviewer #2: Yes

5. Is the manuscript presented in an intelligible fashion and written in standard English?

Reviewer #2: Yes

6. Review Comments to the Author

You may also provide optional suggestions and comments to authors that they might find helpful in planning their study.

Reviewer #2: This is a resubmission. The concerns and comments that were previously raised by reviewers have been satisfactorily addressed by the authors. I have no additional comments. Thank you!

7. PLOS authors have the option to publish the peer review history of their article (what does this mean?). If published, this will include your full peer review and any attached files.

Reviewer #2: **Yes: **LM Ho

---

## [Editor Report · Acceptance letter]

23 Jul 2024

PONE-D-24-18212R1 

PLOS ONE

Dear Dr. Dibaba, 

I'm pleased to inform you that your manuscript has been deemed suitable for publication in PLOS ONE. Congratulations! Your manuscript is now being handed over to our production team.

Kind regards, 

on behalf of

Dr. Ashish Wasudeo Khobragade 

Academic Editor

PLOS ONE